# DOMAIN ADAPTATION FOR STRUCTURED OUTPUT VIA DISENTANGLED PATCH REPRESENTATIONS

## ABSTRACT

Predicting structured outputs such as semantic segmentation relies on expensive per-pixel annotations to learn strong supervised models like convolutional neural networks. However, these models trained on one data domain may not generalize well to other domains unequipped with annotations for model finetuning. To avoid the labor-intensive process of annotation, we develop a domain adaptation method to adapt the source data to the unlabeled target domain. To this end, we propose to learn discriminative feature representations of patches based on label histograms in the source domain, through the construction of a disentangled space. With such representations as guidance, we then use an adversarial learning scheme to push the feature representations in target patches to the closer distributions in source ones. In addition, we show that our framework can integrate a global alignment process with the proposed patch-level alignment and achieve state-of-the-art performance on semantic segmentation. Extensive ablation studies and experiments are conducted on numerous benchmark datasets with various settings, such as synthetic-to-real and cross-city scenarios.

## 1 INTRODUCTION

Recent deep learning-based methods have made significant progress on vision tasks, such as object recognition (Krizhevsky et al., 2012) and semantic segmentation (Long et al., 2015a), relying on large-scale annotations to supervise the learning process. However, for a test domain different from the annotated training data, learned models usually do not generalize well. In such cases, domain adaptation methods have been developed to close the gap between a source domain with annotations and a target domain without labels. Along this line of research, numerous methods have been developed for image classification (Saenko et al., 2010; Ganin & Lempitsky, 2015), but despite recent works on domain adaptation for pixel-level prediction tasks such as semantic segmentation (Hoffman et al., 2016), there still remains significant room for improvement. Yet domain adaptation is a crucial need for pixel-level predictions, as the cost to annotate ground truth is prohibitively expensive. For instance, road-scene images in different cities may have various appearance distributions, while conditions even within the same city may vary significantly over time or weather.

Existing state-of-the-art methods use feature-level (Hoffman et al., 2016) or output space adaptation (Tsai et al., 2018) to align the distributions between the source and target domains using adversarial learning (Goodfellow et al., 2014; Zhu et al., 2017). These approaches usually exploit the global distribution alignment, such as spatial layout, but such global statistics may already differ significantly between two domains due to differences in camera pose or field of view. Figure 1 illustrates one example, where two images share a similar layout, but the corresponding grids do not match well. Such misalignment may introduce an incorrect bias during adaptation. Instead, we consider to match patches that are more likely to be shared across domains regardless of where they are located.

One way to utilize patch-level information is to align their distributions through adversarial learning. However, this is not straightforward since patches may have high variation among each other and there is no guidance for the model to know which patch distributions are close. Motivated by recent advances in learning disentangled representations (Kulkarni et al., 2015; Odena et al., 2017), we adopt a similar approach by considering label histograms of patches as a factor and learn discriminative

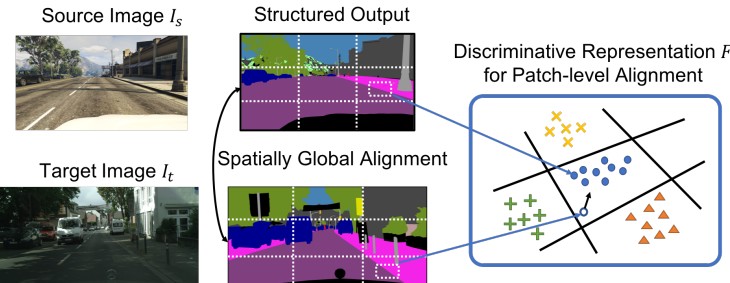

Figure 1: Illustration of the proposed patch-level alignment against the global alignment that considers the spatial relationship between grids. We first learn discriminative representations for source patches (solid symbols) and push a target representation (unfilled symbol) close to the distribution of source ones, regardless of where these patches are located in the image.

representations for patches to relax the high-variation problem among them. Then, we use the learned representations as a bridge to better align patches between source and target domains.

Specifically, we utilize two adversarial modules to align both the global and patch-level distributions between two domains, where the global one is based on the output space adaptation (Tsai et al., 2018), and the patch-based one is achieved through the proposed alignment by learning discriminative representations. To guide the learning process, we first use the pixel-level annotations provided in the source domain and extract the label histogram as a patch-level representation. We then apply K-means clustering to group extracted patch representations into $K$ clusters, whose cluster assignments are then used as the ground truth to train a classifier shared across two domains for transferring a learned discriminative representation of patches from the source to the target domain. Ideally, given the patches in the target domain, they would be classified into one of $K$ categories. However, since there is a domain gap, we further use an adversarial loss to push the feature representations of target patches close to the distribution of the source patches in this clustered space (see Figure 1). Note that our representation learning can be viewed as a kind of disentanglement guided by the label histogram, but is different from existing methods that use pre-defined factors such as object pose (Kulkarni et al., 2015).

In experiments, we follow the domain adaptation setting in (Hoffman et al., 2016) and perform pixel-level road-scene image segmentation. We conduct experiments under various settings, including the synthetic-to-real, i.e., GTA5 (Richter et al., 2016)/SYNTHIA (Ros et al., 2016) to Cityscapes (Cordts et al., 2016) and cross-city, i.e., Cityscapes to Oxford RobotCar (Maddern et al., 2017) scenarios. In addition, we provide extensive ablation studies to validate each component in the proposed framework. By combining global and patch-level alignments, we show that our approach performs favorably against state-of-the-art methods in terms of accuracy and visual quality. We note that the proposed framework is general and could be applicable to other forms of structured outputs such as depth, which will be studied in our future work.

The contributions of this work are as follows. First, we propose a domain adaptation framework for structured output prediction by utilizing global and patch-level adversarial learning modules. Second, we develop a method to learn discriminative representations guided by the label histogram of patches via clustering and show that these representations help the patch-level alignment. Third, we demonstrate that the proposed adaptation method performs favorably against various baselines and state-of-the-art methods on semantic segmentation.

## 2 RELATED WORK

Within the context of this work, we discuss the domain adaptation methods, including image classification and pixel-level prediction tasks. In addition, algorithms that are relevant to learning disentangled representations are discussed in this section.

**Domain Adaptation.** Domain adaptation approaches have been developed for the image classification task via aligning the feature distributions between the source and target domains. Conventional methods use hand-crafted features (Gong et al., 2012; Fernando et al., 2013) to minimize the discrep-

ancy across domains, while recent algorithms utilize deep architectures (Ganin & Lempitsky, 2015; Tzeng et al., 2015) to learn domain-invariant features. One common practice is to adopt the adversarial learning scheme (Ganin et al., 2016) and minimize the Maximum Mean Discrepancy (Long et al., 2015b). A number of variants have been developed via designing different classifiers (Long et al., 2016) and loss functions (Tzeng et al., 2017; 2015). In addition, other recent work aims to enhance feature representations by pixel-level transfer (Bousmalis et al., 2017) and domain separation (Bousmalis et al., 2016).

Compared to the image classification task, domain adaptation for structured pixel-level predictions has not been widely studied. Hoffman et al. (2016) first introduce to tackle the domain adaptation problem on semantic segmentation for road-scene images, e.g., synthetic-to-real images. Similar to the image classification case, they propose to use adversarial networks and align global feature representations across two domains. In addition, a category-specific prior is extracted from the source domain and is transferred to the target distribution as a constraint. However, these priors, e.g., object size and class distribution, may be already inconsistent between two domains. Instead of designing such constraints, the CDA method (Zhang et al., 2017) applies the SVM classifier to capture label distributions on superpixels as the property to train the adapted model on the target domain. Similarly, as proposed in (Chen et al., 2017), a class-wise domain adversarial alignment is performed by assigning pseudo labels to the target data. Moreover, an object prior is extracted from Google Street View to help alignment for static objects.

The above-mentioned domain adaptation methods on structured output all use a global distribution alignment and some class-specific priors to match statistics between two domains. However, such class-level alignment does not preserve the structured information like the patches. In contrast, we propose to learn discriminative representations for patches and use these learned representations to help patch-level alignment. Moreover, our framework does not require additional priors/annotations and the entire network can be trained in an end-to-end fashion. Compared to the recently proposed output space adaptation method (Tsai et al., 2018) that also enables end-to-end training, our algorithm focuses on learning patch-level representations that aid the alignment process.

**Learning Disentangled Representation.**   Learning a latent disentangled space has led to a better understanding for numerous tasks such as facial recognition (Reed et al., 2014), image generation (Chen et al., 2016b; Odena et al., 2017), and view synthesis (Kulkarni et al., 2015; Yang et al., 2015). These approaches use pre-defined factors to learn interpretable representations of the image. Kulkarni et al. (2015) propose to learn graphic codes that are disentangled with respect to various image transformations, e.g., pose and lighting, for rendering 3D images. Similarly, Yang et al. (2015) synthesize 3D objects from a single image via an encoder-decoder architecture that learns latent representations based on the rotation factor. Recently, AC-GAN (Odena et al., 2017) develops a generative adversarial network (GAN) with an auxiliary classifier conditioned on the given factors such as image labels and attributes.

Although these methods present promising results on using the specified factors and learning a disentangled space to help the target task, they focus on handling the data in a single domain. Motivated by this line of research, we propose to learn discriminative representations for patches to help the domain adaptation task. To this end, we take advantages of the available label distributions and naturally utilize them as a disentangled factor, in which our framework does not require to pre-define any factors like conventional methods.

## 3   DOMAIN ADAPTATION FOR STRUCTURED OUTPUT

In this section, we describe our proposed domain adaptation framework for predicting structured outputs, our adversarial learning scheme to align distributions across domains, and the use of discriminative representations for patches to help the alignment.

### 3.1   ALGORITHMIC OVERVIEW

Given the source and target images $I_s, I_t \in \mathbb{R}^{H \times W \times 3}$ and the source labels $Y_s$, our goal is to align the predicted output distribution $O_t$ of the target data with the source distribution $O_s$. As shown in Figure 2(a), we use a loss function for supervised learning on the source data to predict the structured output, and an adversarial loss is adopted to align the global distribution. Based on this

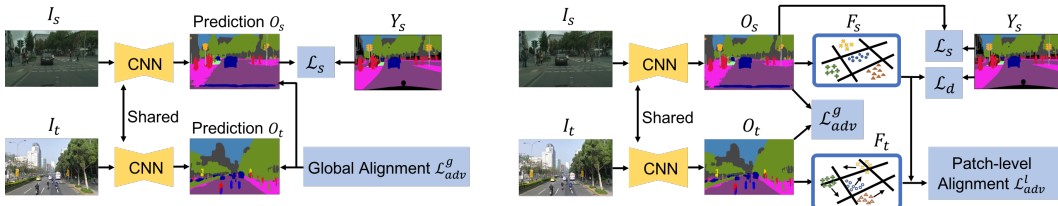

(a) Global alignment (baseline model)  (b) Our framework with patch-level alignment

Figure 2: Overview of the proposed framework. (a) A baseline model that utilizes global alignment in (Tsai et al., 2018). (b) Our algorithm combines global and patch-level alignments in a general framework that better preserves local structure. Note that $O_s, O_t \in (0,1)^{H \times W \times C}$ are distributions and the figures used here are only for illustration purposes.

baseline model, we further incorporate a classification loss in a clustered space to learn patch-level discriminative representations $F_s$ from the source output distribution $O_s$, shown in Figure 2(b). For target data, we employ another adversarial loss to align the patch-level distributions between $F_s$ and $F_t$, where the goal is to push $F_t$ to be closer to the distribution of $F_s$.

**Objective Function.**  As described in Figure 2(b), we formulate the adaptation task as composed of the following loss functions:

$$\mathcal{L}_{total}(I_s, I_t, Y_s, \Gamma(Y_s)) = \mathcal{L}_s + \lambda_d \mathcal{L}_d + \lambda_{adv}^g \mathcal{L}_{adv}^g + \lambda_{adv}^l \mathcal{L}_{adv}^l, \tag{1}$$

where $\mathcal{L}_s$ and $\mathcal{L}_d$ are supervised loss functions for learning the structured prediction and the discriminative representation on source data, respectively, while $\Gamma$ denotes the clustering process on the ground truth label distribution. To align the target distribution, we utilize global and patch-level adversarial loss functions, which are denoted as $\mathcal{L}_{adv}^g$ and $\mathcal{L}_{adv}^l$, respectively. Here, $\lambda$'s are the weights for different loss functions. The following sections describe details of the baseline model and the proposed framework. Figure 3 shows the main components and loss functions of our method.

## 3.2 Global Alignment with Adversarial Learning

We first adopt a baseline model that consists of a supervised cross-entropy loss $\mathcal{L}_s$ and an output space adaptation module using $\mathcal{L}_{adv}^g$ for global alignment as shown in Figure 2(a). The loss $\mathcal{L}_s$ can be optimized by a fully-convolutional network $\mathbf{G}$ that predicts the structured output with the loss summed over the spatial map indexed with $h, w$ and the number of categories $C$:

$$\mathcal{L}_s(I_s, Y_s; \mathbf{G}) = -\sum_{h,w} \sum_{c \in C} Y_s^{(h,w,c)} \log(O_s^{(h,w,c)}), \tag{2}$$

where $O_s = \mathbf{G}(I_s) \in (0,1)$ is the predicted output distribution through the softmax function and is up-sampled to the size of the input image. Here, we will use the same $h$ and $w$ as the index for all the formulations. For the adversarial loss $\mathcal{L}_{adv}^g$, we follow the practice of GAN training by optimizing $\mathbf{G}$ and a discriminator $\mathbf{D}_g$ that performs the binary classification to distinguish whether the output prediction is from the source image or the target one.

$$\mathcal{L}_{adv}^g(I_s, I_t; \mathbf{G}, \mathbf{D}_g) = \sum_{h,w} \mathbb{E}[\log \mathbf{D}_g(O_s)^{(h,w,1)}] + \mathbb{E}[\log(1 - \mathbf{D}_g(O_t)^{(h,w,1)})]. \tag{3}$$

Then we optimize the following min-max problem for $\mathbf{G}$ and $\mathbf{D}_g$, with inputs to the functions dropped for simplicity:

$$\min_{\mathbf{G}} \max_{\mathbf{D}_g} \mathcal{L}_s(\mathbf{G}) + \lambda_{adv}^g \mathcal{L}_{adv}^g(\mathbf{G}, \mathbf{D}_g). \tag{4}$$

## 3.3 Patch-level Alignment with Discriminative Representations

Figure 1 shows that we may find transferable structured output representations shared across source and target images from smaller patches rather than from the entire image or larger grids. Based on this observation, we propose to perform a patch-level domain alignment. Specifically, rather than naively aligning the distributions of all patches between two domains, we first perform clustering

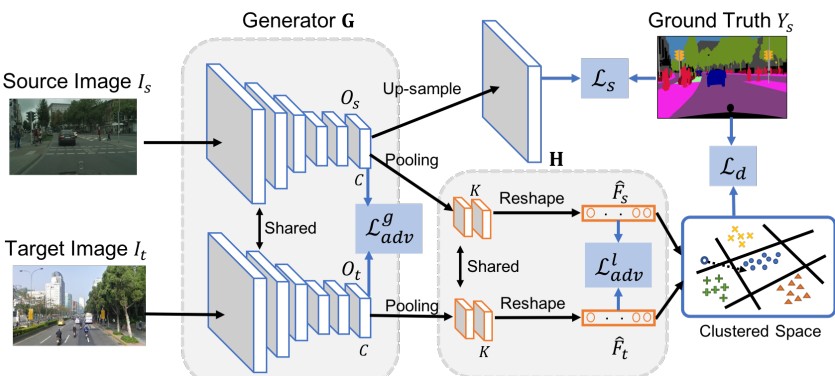

Figure 3: The proposed network architecture that consists of a generator **G** and a categorization module **H** for learning discriminative patch representations. In the clustered space, solid symbols denote source representations and unfilled ones are target representations pulled to the source distribution.

on patches from the source-domain examples using ground truth segmentation labels to construct a set of prototypical patch patterns. Then, we let patches from the target domain adapt to this disentangled (clustered) space of source patch representations by guiding them to select the closest cluster regardless of the spatial location via adversarial objective. In the following, we describe details of the proposed patch-level alignment.

**Learning Discriminative Representations.** In order to learn a disentangled space, class labels (Salimans et al., 2016) or pre-defined factors (Odena et al., 2017) are usually provided as supervisory signals. However, it is non-trivial to assign some sort of class membership to individual patches of an image. One may apply unsupervised clustering of image patches using pixel representations, but it is unclear whether the constructed clustering would separate patches in a semantically meaningful way. In this work, we take advantage of already available per-pixel annotations in the source domain to construct semantically disentangled space of patch representations.

To achieve this, we use label histograms for patches as the disentangled factor. We first randomly sample patches from source images, use a 2-by-2 grid on patches to extract spatial label histograms, and concatenate them into a vector, where each histogram is a $2 \cdot 2 \cdot C$ dimensional vector. Second, we apply K-means clustering on these histograms, whereby the label for any patch can be assigned as the cluster center with the closest distance on the histogram.

To incorporate this clustered space during training the network **G** on source data, we add a classification module **H** after the predicted output $O_s$, to simulate the procedure of constructing the label histogram and learn a discriminative representation. We denote the learned representation as $F_s = \mathbf{H}(\mathbf{G}(I_s)) \in (0,1)^{U \times V \times K}$ through the softmax function, where $K$ is the number of clusters. Here, each data point on the spatial map $F_s$ corresponds to a patch of the input image, and we obtain the group label $\Gamma(Y_s)$ for each patch accordingly. Then the learning process to construct the clustered space can be formulated as a cross-entropy loss:

$$\mathcal{L}_d(I_s, \Gamma(Y_s); \mathbf{G}, \mathbf{H}) = -\sum_{u,v} \sum_{k \in K} \Gamma(Y_s)^{(u,v,k)} \log(F_s^{(u,v,k)}). \tag{5}$$

**Patch-level Adversarial Alignment.** The ensuing task is to align the representations of target patches to the clustered space constructed in the source domain. To this end, we utilize another adversarial loss between $F_s$ and $F_t$, where $F_t$ is generated in the same way as described above. Our goal is to align patches regardless of where they are located in the image, that is, without the spatial and neighborhood supports. Thus, we reshape $F$ by concatenating the $K$-dimensional vectors along the spatial map, which results in $U \cdot V$ independent data points. We note that a similar effect can be achieved by using a convolution layer with a proper stride and kernel size. We denote this reshaped data as $\hat{F}$ and formulate the adversarial objective:

$$\mathcal{L}_{adv}^l(I_s, I_t; \mathbf{G}, \mathbf{H}, \mathbf{D}_l) = \sum_{u,v} \mathbb{E}[\log \mathbf{D}_l(\hat{F}_s)^{(u,v,1)}] + \mathbb{E}[\log(1 - \mathbf{D}_l(\hat{F}_t)^{(u,v,1)})], \tag{6}$$

where $\mathbf{D}_l$ is the discriminator to classify whether the feature representation $\hat{F}$ is from the source or the target domain. Finally, we integrate (5) and (6) into the min-max problem in (4):

$$\min_{\mathbf{G},\mathbf{H}} \max_{\mathbf{D}_g,\mathbf{D}_l} \mathcal{L}_s(\mathbf{G}) + \lambda_d \mathcal{L}_d(\mathbf{G},\mathbf{H}) + \lambda_{adv}^g \mathcal{L}_{adv}^g(\mathbf{G},\mathbf{D}_g) + \lambda_{adv}^l \mathcal{L}_{adv}^l(\mathbf{G},\mathbf{H},\mathbf{D}_l). \qquad (7)$$

## 3.4 NETWORK OPTIMIZATION

Following the standard procedure for training a GAN (Goodfellow et al., 2014), we alternate the optimization between three steps: 1) update the discriminator $\mathbf{D}_g$, 2) update the discriminator $\mathbf{D}_l$, and 3) update the network $\mathbf{G}$ and $\mathbf{H}$ while fixing the discriminators.

**Update the Discriminator $\mathbf{D}_g$.** We train the discriminator $\mathbf{D}_g$ to distinguish between the source output distribution (labeled as 1) and the target distribution (labeled as 0). The maximization problem on $\mathbf{D}_g$ in (7) is equivalent to minimizing the binary cross-entropy loss:

$$\mathcal{L}_D^g(O_s, O_t; \mathbf{D}_g) = -\sum_{h,w} \log(\mathbf{D}_g(O_s)^{(h,w,1)}) + \log(1 - \mathbf{D}_g(O_t)^{(h,w,1)}). \qquad (8)$$

**Update the Discriminator $\mathbf{D}_l$.** Similarly, we train the discriminator $\mathbf{D}_l$ to classify whether the feature representation $\hat{F}$ is from the source or the target domain:

$$\mathcal{L}_D^l(\hat{F}_s, \hat{F}_t; \mathbf{D}_l) = -\sum_{u,v} \log(\mathbf{D}_l(\hat{F}_s)^{(u,v,1)}) + \log(1 - \mathbf{D}_l(\hat{F}_t)^{(u,v,1)}). \qquad (9)$$

**Update the Network G and H.** The goal of this step is to push the target distribution closer to the source distribution using the optimized $\mathbf{D}_g$ and $\mathbf{D}_l$, while maintaining good performance on the main tasks using $\mathbf{G}$ and $\mathbf{H}$. As a result, the minimization problem in (7) is the combination of two supervised loss functions, namely, (2) and (5), with two adversarial loss functions, where the adversarial ones can be expressed as binary cross-entropy loss functions that assign the source label to the target distribution:

$$\mathcal{L}_{total} = \mathcal{L}_s + \lambda_d \mathcal{L}_d - \lambda_{adv}^g \sum_{h,w} \log(\mathbf{D}_g(O_t)^{(h,w,1)}) - \lambda_{adv}^l \sum_{u,v} \log(\mathbf{D}_l(\hat{F}_t)^{(u,v,1)}). \qquad (10)$$

We note that updating $\mathbf{H}$ would also update $\mathbf{G}$ through back-propagation, and thus the feature representations are enhanced in $\mathbf{G}$. In addition, we only require $\mathbf{G}$ during the testing phase, so that runtime is unaffected compared to the baseline approach.

## 3.5 NETWORK ARCHITECTURE AND IMPLEMENTATION DETAILS

**Discriminator.** For the discriminator $\mathbf{D}_g$ using a spatial map $O$ as the input, we adopt an architecture similar to (Radford et al., 2016) but use fully-convolutional layers. It contains 5 convolution layers with kernel size $4 \times 4$, stride 2 and channel numbers $\{64, 128, 256, 512, 1\}$. In addition, a leaky ReLU activation (Maas et al., 2013) is added after each convolution layer, except the last layer. For the discriminator $\mathbf{D}_l$, input data is a $K$-dimensional vector and we utilize 3 fully-connected layers similar to (Tzeng et al., 2017), with leaky ReLU activation and channel numbers $\{256, 512, 1\}$.

**Generator.** The generator consists of the network $\mathbf{G}$ with a categorization module $\mathbf{H}$. For a fair comparison, we follow the framework used in (Tsai et al., 2018) that adopts DeepLab-v2 (Chen et al., 2016a) with the ResNet-101 architecture (He et al., 2016) pre-trained on ImageNet (Deng et al., 2009) as our baseline network $\mathbf{G}$. To add the module $\mathbf{H}$ on the output prediction $O$, we first use an adaptive average pooling layer to generate a spatial map, where each data point on the map has a desired receptive field corresponding to the size of extracted patches. Then this pooled map is fed into two convolution layers and a feature map $F$ is produced with the channel number $K$. Figure 3 illustrates the main components of the proposed architecture.

**Implementation Details.** We implement the proposed framework using the PyTorch toolbox on a single Titan X GPU with 12 GB memory. To train the discriminators, we use the Adam optimizer (Kingma & Ba, 2015) with initial learning rate of $10^{-4}$ and momentums set as 0.9 and 0.99. For learning the generator, we use the Stochastic Gradient Descent (SGD) solver where the momentum is 0.9, the weight decay is $5 \times 10^{-4}$ and the initial learning rate is $2.5 \times 10^{-4}$. For all

Table 1: Ablation study on GTA5-to-Cityscapes using the ResNet-101 network. We also show the corresponding loss functions used in each setting.

| | GTA5 → Cityscapes | | | |
|---|---|---|---|---|
| | **Without Adaptation** $\mathcal{L}_s$ | **Disentanglement** $\mathcal{L}_s + \mathcal{L}_d$ | **Global Alignment** $\mathcal{L}_s + \mathcal{L}_{adv}^g$ | **Patch-level Alignment** $\mathcal{L}_s + \mathcal{L}_d + \mathcal{L}_{adv}^l$ |
| mIoU | 36.6 | 38.8 | 41.4 | 41.3 |
| | **Without $\mathcal{L}_d$** $\mathcal{L}_s + \mathcal{L}_{adv}^g + \mathcal{L}_{adv}^l$ | **Without $\mathcal{L}_{adv}^l$** $\mathcal{L}_s + \mathcal{L}_d + \mathcal{L}_{adv}^g$ | **Without Reshaped $\hat{F}$** $\mathcal{L}_s + \mathcal{L}_d + \mathcal{L}_{adv}^g + \mathcal{L}_{adv}^l$ | **Ours (final)** $\mathcal{L}_s + \mathcal{L}_d + \mathcal{L}_{adv}^g + \mathcal{L}_{adv}^l$ |
| mIoU | 41.3 | 41.7 | 40.8 | 43.2 |

the networks, we decrease the learning rates using the polynomial decay with a power of 0.9, as described in (Chen et al., 2016a). During training, we use $\lambda_d = 0.01$, $\lambda_{adv}^g = \lambda_{adv}^l = 0.0005$ and $K = 50$ for all the experiments. Note that we first train the model only using the loss $\mathcal{L}_s$ for 10K iterations to avoid initially noisy predictions and then train the network using all the loss functions for 100K iterations. More details of the hyper-parameters such as image and patch sizes are provided in the appendix.

## 4 EXPERIMENTAL RESULTS

We evaluate the proposed framework for domain adaptation on semantic segmentation. We first conduct an extensive ablation study to validate each component in the algorithm on the GTA5-to-Cityscapes (synthetic-to-real) scenario. Second, we show that our method performs favorably against state-of-the-art approaches on numerous benchmark datasets and settings.

### 4.1 EVALUATED DATASETS AND METRIC

We evaluate our domain adaptation method on semantic segmentation under various settings, including synthetic-to-real and cross-city scenarios. First, we adapt the synthetic GTA5 (Richter et al., 2016) dataset to the Cityscapes (Cordts et al., 2016) dataset that contains real road-scene images. Similarly, we use the SYNTHIA (Ros et al., 2016) dataset with a larger domain gap compared to Cityscapes images. For the above experiments, we follow (Hoffman et al., 2016) to split the training and test sets. To overcome the realistic case when two domains are in different cities under various weather conditions, we adapt Cityscapes with sunny images to the Oxford RobotCar (Maddern et al., 2017) dataset that contains rainy scenes. We manually select 10 sequences in the Oxford RobotCar dataset annotated with the rainy tag, in which we randomly split them into 7 sequences for training and 3 for testing. We sequentially sample 895 images as training images and annotate 271 images with per-pixel semantic segmentation ground truth as the test set for evaluation. The annotated ground truth will be made publicly available. For all the experiments, intersection-over-union (IoU) ratio is used as the metric to evaluate different methods.

### 4.2 ABLATION STUDY

In Table 1, we conduct an ablation study on the GTA5-to-Cityscapes scenario to understand the impact of different loss functions and design choices in the proposed framework.

**Loss Functions.** In the first row of Table 1, we show different steps of the proposed method, including disentanglement, global alignment, and patch-level alignment. Interestingly, we find that adding disentanglement without any alignments ($\mathcal{L}_s + \mathcal{L}_d$) also improves the performance (from 36.6% to 38.8%), which demonstrates that the learned feature representation enhances the discrimination and generalization ability. Finally, as shown in the last result of the second row, our method that combines both the global and patch-level alignments achieve the highest IoU as 43.2%.

**Impact on $\mathcal{L}_d$ and $\mathcal{L}_{adv}^l$.** In the first two results of the second row, we conduct experiments to validate the effectiveness of our patch-level alignment. We show that both losses, $\mathcal{L}_d$ and $\mathcal{L}_{adv}^l$, are necessary to assist this alignment process. Removing either of them will result in performance loss, i.e., 1.9% and 1.5% lower than our final result. The reason behind this is that, $\mathcal{L}_d$ is to construct a clustered space so that $\mathcal{L}_{adv}^l$ can then effectively perform patch-level alignment in this space.

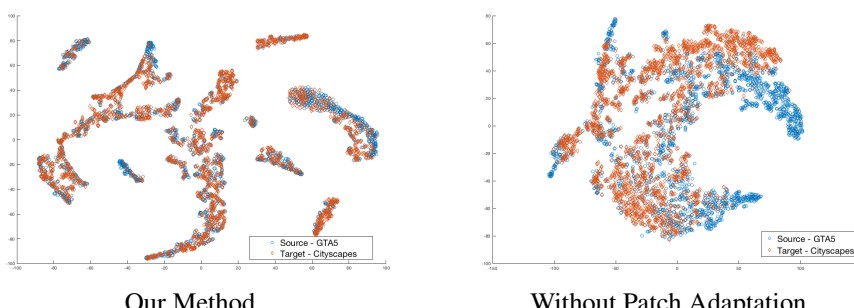

Our Method                                    Without Patch Adaptation

Figure 4: Visualization of patch-level feature representations via t-SNE. In each figure, two thousand patches are sampled from each source and target domain.

Table 2: Results of adapting GTA5 to Cityscapes. In the first and second groups, VGG-16 and ResNet-101 base networks are adopted, respectively.

| Method | road | sidewalk | building | wall | fence | pole | light | sign | veg | terrain | sky | person | rider | car | truck | bus | train | mbike | bike | mIoU |
|---|---|---|---|---|---|---|---|---|---|---|---|---|---|---|---|---|---|---|---|---|
| | | | | | | | | | GTA5 → Cityscapes | | | | | | | | | | | |
| Hoffman et al. (2016) | 70.4 | 32.4 | 62.1 | 14.9 | 5.4 | 10.9 | 14.2 | 2.7 | 79.2 | 21.3 | 64.6 | 44.1 | 4.2 | 70.4 | 8.0 | 7.3 | 0.0 | 3.5 | 0.0 | 27.1 |
| Zhang et al. (2017) | 74.9 | 22.0 | 71.7 | 6.0 | 11.9 | 8.4 | 16.3 | 11.1 | 75.7 | 13.3 | 66.5 | 38.0 | **9.3** | 55.2 | 18.8 | 18.9 | 0.0 | **16.8** | **14.6** | 28.9 |
| Hoffman et al. (2018) | 83.5 | **38.3** | 76.4 | 20.6 | 16.5 | 22.2 | **26.2** | **21.9** | 80.4 | 28.7 | 65.7 | **49.4** | 4.2 | 74.6 | 16.0 | 26.6 | 2.0 | 8.0 | 0.0 | 34.8 |
| Tsai et al. (2018) | **87.3** | 29.8 | 78.6 | 21.1 | 18.2 | 22.5 | 21.5 | 11.0 | 79.7 | 29.6 | 71.3 | 46.8 | 6.5 | 80.1 | 23.0 | **26.9** | 0.0 | 10.6 | 0.3 | 35.0 |
| Ours (VGG-16) | 87.0 | 31.5 | **79.4** | **30.5** | **21.4** | **24.5** | 19.6 | 10.4 | **80.6** | **30.8** | **72.1** | 48.8 | 6.6 | **81.4** | **23.5** | 14.7 | **8.4** | 16.4 | 1.7 | **36.3** |
| Without Adaptation | 75.8 | 16.8 | 77.2 | 12.5 | **21.0** | 25.5 | 30.1 | 20.1 | 81.3 | 24.6 | 70.3 | 53.8 | 26.4 | 49.9 | 17.2 | 25.9 | 6.5 | 25.3 | **36.0** | 36.6 |
| Tsai et al. (2018) (Feature) | 83.7 | 27.6 | 75.5 | 20.3 | 19.9 | 27.4 | 28.3 | **27.4** | 79.0 | 28.4 | 70.1 | 55.1 | 20.2 | 72.9 | 22.5 | 35.7 | **8.3** | 20.6 | 23.0 | 39.3 |
| Tsai et al. (2018) (Output) | 86.5 | 25.9 | 79.8 | 22.1 | 20.0 | 23.6 | **33.1** | 21.8 | 81.8 | 25.9 | 75.9 | 57.3 | 26.2 | 76.3 | 29.8 | 32.1 | 7.2 | **29.5** | 32.5 | 41.4 |
| Ours (ResNet-101) | **89.2** | **38.4** | 80.4 | 24.4 | 21.0 | 27.7 | 32.9 | 16.1 | **83.1** | **34.1** | **77.8** | 57.4 | **27.6** | 78.6 | 31.2 | 40.2 | 4.7 | 27.6 | 27.6 | **43.2** |

**Without Reshaped $\hat{F}$.** In the module **H** that transforms the output distribution to the clustered space, the features are reshaped as independent data points $\hat{F}$ to remove the spatial relationship and are then used as the input to the discriminator $\mathbf{D}_l$. To validate the usefulness, we show that without the reshaping process, the performance drops 2.4% in IoU. This result matches our assumption that patches with similar representations should be aligned regardless of their locations.

**Visualization of Feature Representations.** In Figure 4, we show the t-SNE visualization (van der Maaten & Hinton, 2008) of the patch-level features in the clustered space of our method and compare with the one without patch-level adaptation. The result shows that with adaptation in the clustered space, the features are embedded into groups and the source/target representations overlap to each other well. Example patch visualizations are provided in the appendix.

### 4.3 COMPARISONS WITH STATE-OF-THE-ART METHODS

In this section, we compare the proposed method with state-of-the-art algorithms under various scenarios, including synthetic-to-real and cross-city cases.

**Synthetic-to-real Case.** We first present experimental results for adapting GTA5 to Cityscapes in Table 2. The methods in the upper group adopt the VGG-16 architecture as the base network and we show that our approach performs favorably against state-of-the-art adaptations via feature (Hoffman et al., 2016; Zhang et al., 2017), pixel-level (Hoffman et al., 2018), and output space (Tsai et al., 2018) alignments. In the bottom group, we further utilize the stronger ResNet-101 base network and compare our result with (Tsai et al., 2018) under two settings, i.e., feature and output space adaptations. We show that the proposed method improves the IoU with a gain of 1.8% and achieves the best IoU on 14 out of the 19 categories. In Table 3, we show results for adapting SYNTHIA to Cityscapes and similar improvements are observed comparing with state-of-the-art methods. In addition, we shows visual comparisons in Figure 5 and more results are presented in the appendix.

**Cross-city Case.** Adapting between real images across different cities and conditions is an important scenario for practical applications. We choose a challenge case where the weather condition is different (i.e., sunny v.s rainy) in two cities by adapting Cityscapes to Oxford RobotCar. The proposed

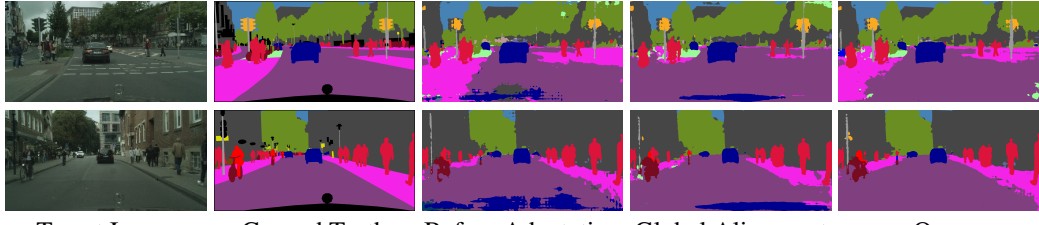

| Target Image | Ground Truth | Before Adaptation | Global Alignment | Ours |

Figure 5: Example results for GTA5-to-Cityscapes. Our method often generates the segmentation with more details (e.g., sidewalk and pole) while producing less noisy regions.

Table 3: Results of adapting SYNTHIA to Cityscapes. In the first and second groups, VGG-16 and ResNet-101 base networks are adopted, respectively.

| Method | road | sidewalk | building | light | sign | veg | sky | person | rider | car | bus | mbike | bike | mIoU |
|---|---|---|---|---|---|---|---|---|---|---|---|---|---|---|
| | | | | | SYNTHIA → Cityscapes | | | | | | | | | |
| Hoffman et al. (2016) | 11.5 | 19.6 | 30.8 | 0.1 | **11.7** | 42.3 | 68.7 | 51.2 | 3.8 | 54.0 | 3.2 | 0.2 | 0.6 | 22.9 |
| Zhang et al. (2017) | 65.2 | 26.1 | 74.9 | **3.7** | 3.0 | 76.1 | 70.6 | **47.1** | 8.2 | 43.2 | **20.7** | 0.7 | **13.1** | 34.8 |
| Chen et al. (2017) | 62.7 | 25.6 | **78.3** | 1.2 | 5.4 | **81.3** | **81.0** | 37.4 | 6.4 | 63.5 | 16.1 | 1.2 | 4.6 | 35.7 |
| Tsai et al. (2018) | **78.9** | **29.2** | 75.5 | 0.1 | 4.8 | 72.6 | 76.7 | 43.4 | 8.8 | 71.1 | 16.0 | 3.6 | 8.4 | 37.6 |
| Ours (VGG-16) | 76.3 | 29.0 | 75.3 | 0.8 | 6.1 | 72.9 | 78.7 | 44.1 | **9.5** | 71.5 | 18.5 | **4.4** | 8.7 | **38.1** |
| Without Adaptation | 55.6 | 23.8 | 74.6 | 6.1 | **12.1** | 74.8 | 79.0 | **55.3** | 19.1 | 39.6 | 23.3 | 13.7 | 25.0 | 38.6 |
| Tsai et al. (2018) (Feature) | 62.4 | 21.9 | 76.3 | **11.7** | 11.4 | 75.3 | 80.9 | 53.7 | 18.5 | 59.7 | 13.7 | 20.6 | 24.0 | 40.8 |
| Tsai et al. (2018) (Output) | 79.2 | 37.2 | 78.8 | 9.9 | 10.5 | **78.2** | 80.5 | 53.5 | 19.6 | 67.0 | 29.5 | **21.6** | 31.3 | 45.9 |
| Ours (ResNet-101) | **82.2** | **39.4** | **79.4** | 6.5 | 10.8 | 77.8 | **82.0** | 54.9 | **21.1** | **67.7** | **30.7** | 17.8 | **32.2** | **46.3** |

framework achieves a mean IoU of 65.0% averaged on 10 categories, significantly improving the model without adaptation by 8.8%. To compare with the state-of-the-art method (Tsai et al., 2018), we run the authors' released code and obtain a mean IoU of 63.6%, which is 1.4% lower than the proposed method. Further results and comparisons are provided in the appendix.

## 5 CONCLUSIONS

In this paper, we present a domain adaptation method for structured output via a general framework that combines global and patch-level alignments. The global alignment is achieved by the output space adaptation, while the patch-level one is performed via learning discriminative representations of patches across domains. To learn such patch-level representations, we propose to construct a clustered space of the source patches and adopt an adversarial learning scheme to push the target patch distributions closer to the source ones. We conduct extensive ablation study and experiments to validate the effectiveness of the proposed method under numerous challenges on semantic segmentation, including synthetic-to-real and cross-city scenarios, and show that our approach performs favorably against existing algorithms.

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

## A    TRAINING DETAILS

To train the model in an end-to-end manner, we randomly sample one image from each of the source and target domain (i.e., batch size as 1) in a training iteration. Then we follow the optimization strategy as described in Section 3.4 of the paper. Table 4 shows the image and patch sizes during training and testing. Note that, the aspect ratio of the image is always maintained (i.e., no cropping) and then the image is down-sampled to the size as in the table.

Table 4:  Image and patch sizes for training and testing.

| Dataset | Cityscapes | GTA5 | SYNTHIA | Oxford RobotCar |
|---|---|---|---|---|
| Patch size for training | $32 \times 64$ | $36 \times 64$ | $38 \times 64$ | - |
| Image size for training | $512 \times 1024$ | $720 \times 1280$ | $760 \times 1280$ | $960 \times 1280$ |
| Image size for testing | $512 \times 1024$ | - | - | $960 \times 1280$ |

## B    RELATION TO ENTROPY MINIMIZATION

Entropy minimization (Grandvalet & Bengio, 2004) can be used as a loss in our model to push the target feature representation $F_t$ to one of the source clusters. To add this regularization, we replace the adversarial loss on the patch level with an entropy loss as in (Long et al., 2016), i.e., $\mathcal{L}_s + \mathcal{L}_d + \mathcal{L}_{adv}^g + \mathcal{L}_{en}^l$, where $\mathcal{L}_{en}^l = \sum_{u,v} \sum_k H(\sigma(\hat{F}_t/\tau))^{(u,v,k)}$, $H$ is the information entropy function, $\sigma$ is the softmax function, and $\tau$ is the temperature of the softmax. The model with adding this entropy regularization achieves the IoU as 41.9%, that is lower than the proposed patch-level adversarial alignment as 43.2%. The reason is that, different from the entropy minimization approach that does not use the source distribution as the guidance, our model learns discriminative representations for the target patches by pushing them closer to the source distribution in the clustered space guided by the label histogram.

## C    VISUALIZATION OF PATCH-LEVEL REPRESENTATIONS

In Figure 6, we show example patches from the source and target domains corresponding to the t-SNE visualization. For each group in the clustered space via t-SNE, we show that source and target patches share high similarity between each other, which demonstrates the effectiveness of the proposed patch-level alignment.

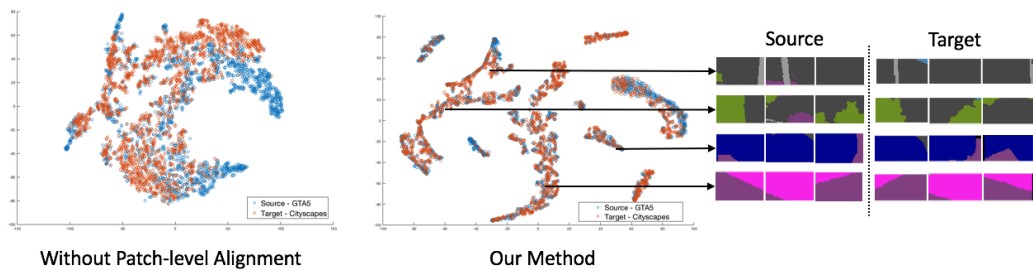

Figure 6: Visualization of patch-level representations. We first show feature representations via t-SNE of our method and compare with the one without the proposed patch-level alignment. In addition, we show patch examples in the clustered space. In each group, patches are similar in appearance between the source and target domains.

## D    RESULT OF ADAPTING CITYSCAPES TO OXFORD ROBOTCAR

In Table 5, we present the complete result for adapting Cityscapes (sunny condition) to Oxford RobotCar (rainy scene). We compare the proposed method with the model without adaptation and the output space adaptation approach (Tsai et al., 2018). More qualitative results are provided in Figure 7 and 8.

Table 5: Results of adapting Cityscapes to Oxford RobotCar.

| Method | road | sidewalk | building | light | sign | veg | sky | person | automobile | two-wheel | mIoU |
|---|---|---|---|---|---|---|---|---|---|---|---|
| | Cityscapes → Oxford RobotCar | | | | | | | | | | |
| Without Adaptation | 79.2 | 49.3 | 73.1 | 55.6 | 37.3 | 4.5 | 36.1 | 54.0 | 81.3 | 49.7 | 56.2 |
| Tsai et al. (2018) | 95.1 | 64.0 | 75.7 | 61.3 | 35.5 | 10.5 | 63.9 | 58.1 | 84.6 | 57.0 | 63.6 |
| Ours | 94.9 | 64.4 | 82.8 | 62.3 | 35.2 | 8.7 | 76.4 | 57.4 | 85.0 | 57.5 | 65.0 |

## E    QUALITATIVE COMPARISONS

We provide more visual comparisons for GTA5-to-Cityscapes and SYNTHIA-to-Cityscapes scenarios from Figure 9 to Figure 11. In each row, we present the results of the model without adaptation, output space adaptation (Tsai et al., 2018), and the proposed method. We show that our approach often yields better segmentation outputs with more details and produces less noisy regions.

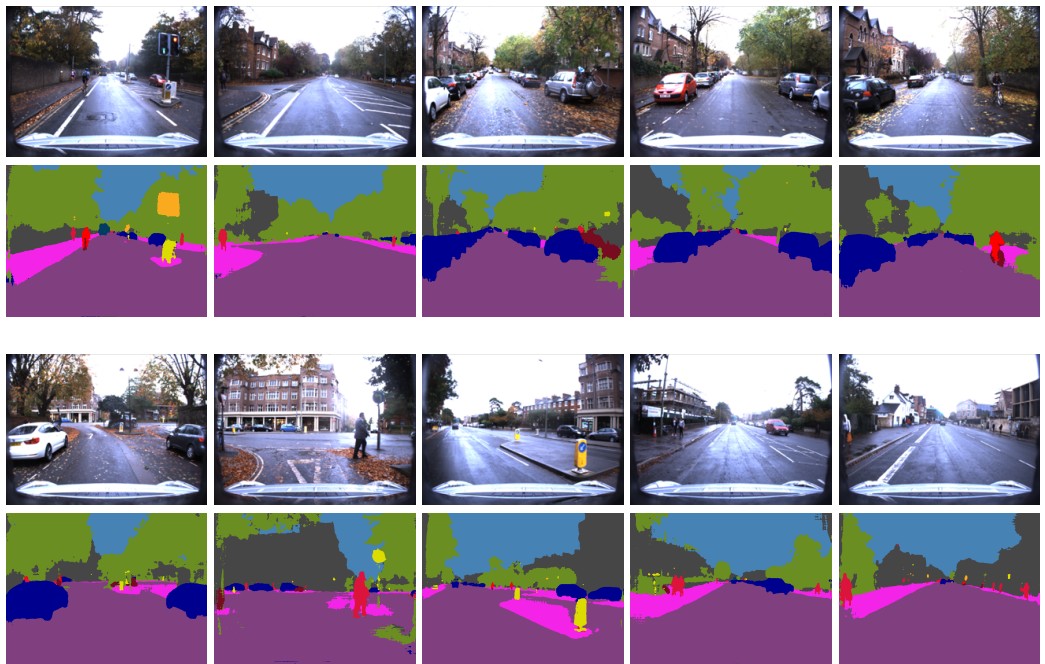

Figure 7: Example results of adapted segmentation for the Cityscapes-to-OxfordRobotCar setting. We sequentially show images in a video and their adapted segmentations generated by our method.

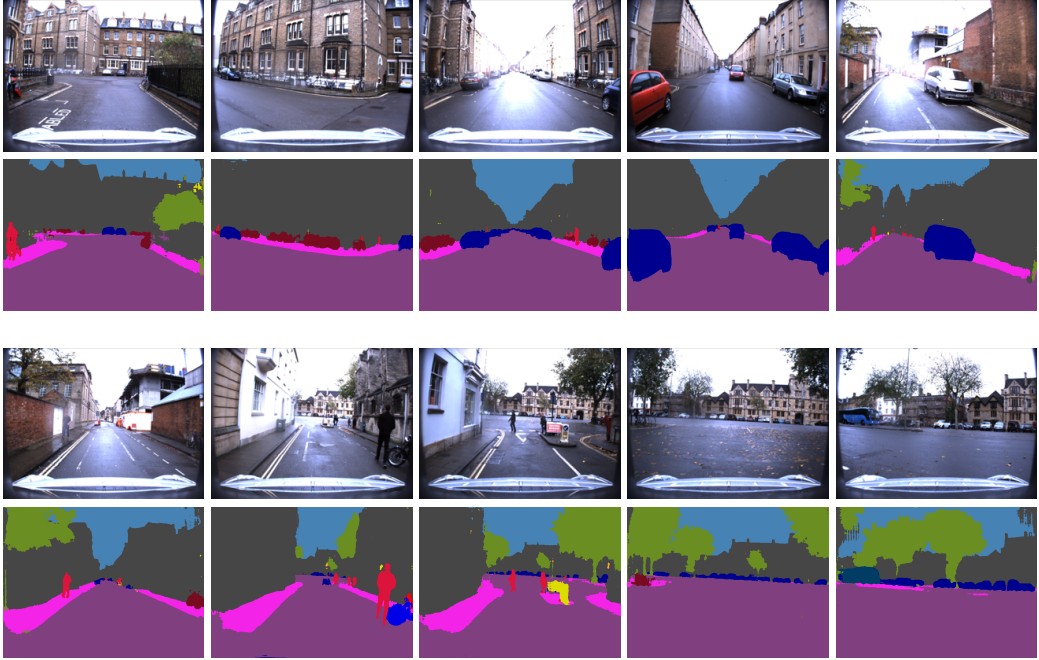

Figure 8: Example results of adapted segmentation for the Cityscapes-to-OxfordRobotCar setting. We sequentially show images in a video and their adapted segmentations generated by our method.

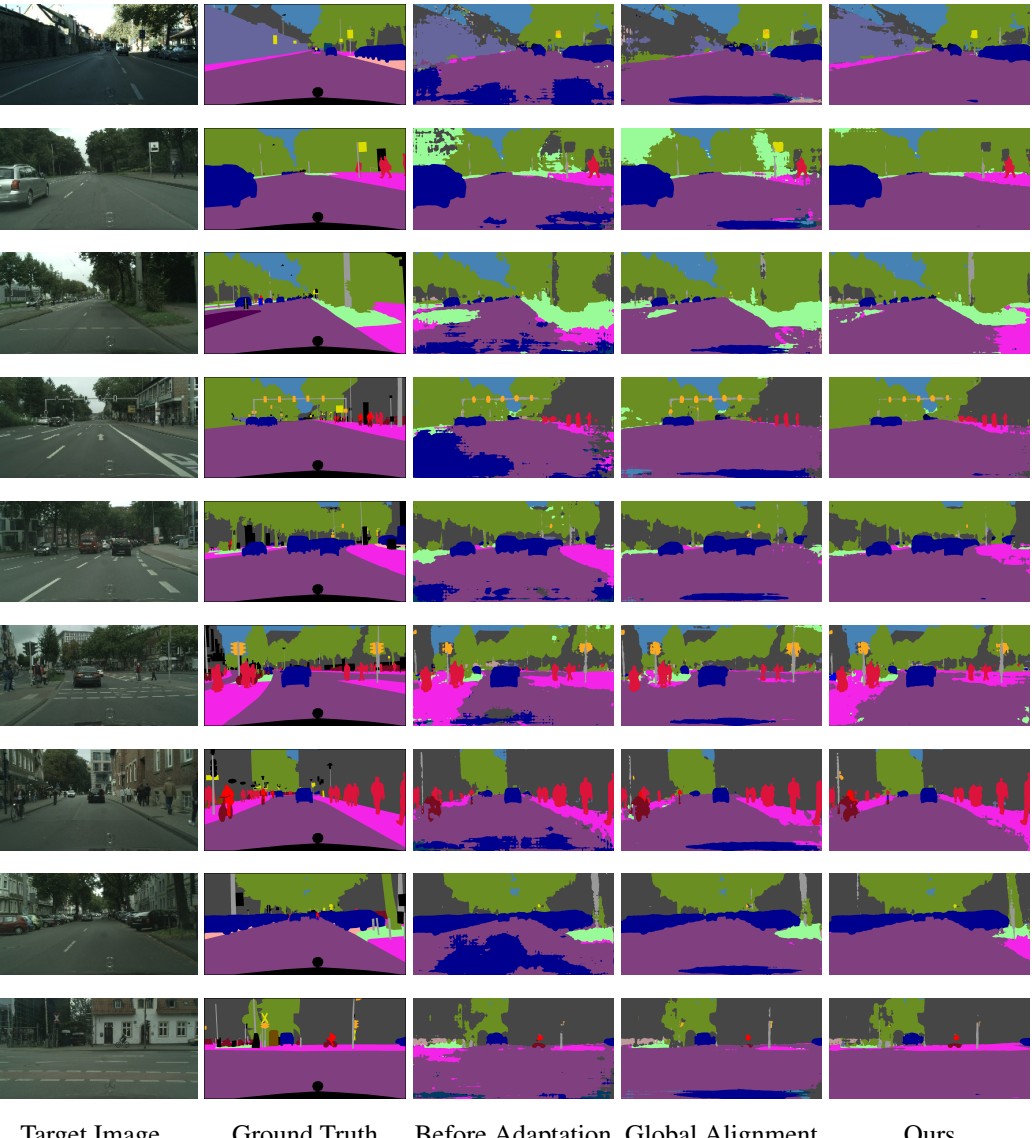

Target Image    Ground Truth    Before Adaptation    Global Alignment    Ours

Figure 9: Example results of adapted segmentation for the GTA5-to-Cityscapes setting. For each target image, we show results before adaptation, output space adaptation Tsai et al. (2018), and the proposed method.

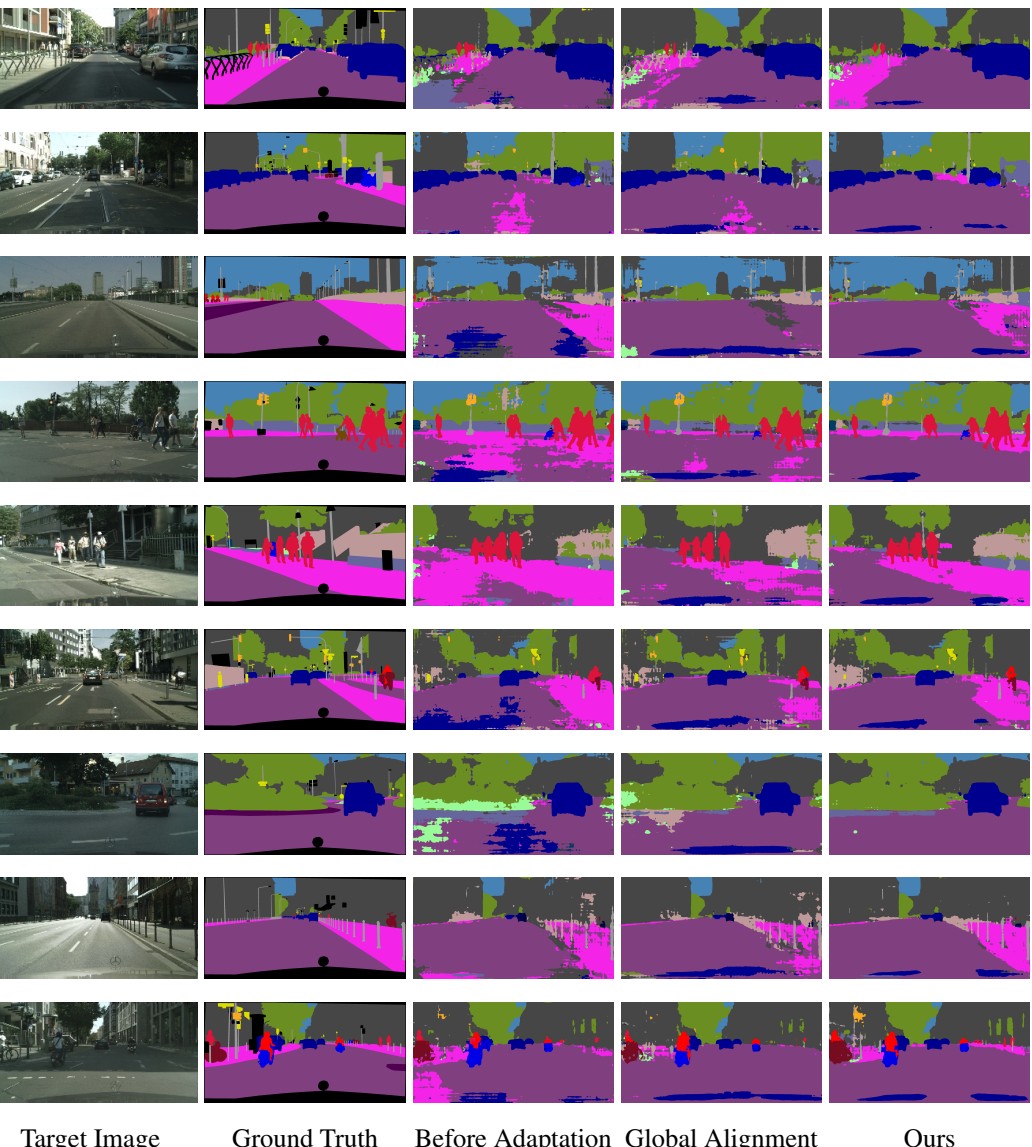

Target Image    Ground Truth    Before Adaptation  Global Alignment    Ours

Figure 10: Example results of adapted segmentation for the GTA5-to-Cityscapes setting. For each target image, we show results before adaptation, output space adaptation Tsai et al. (2018), and the proposed method.

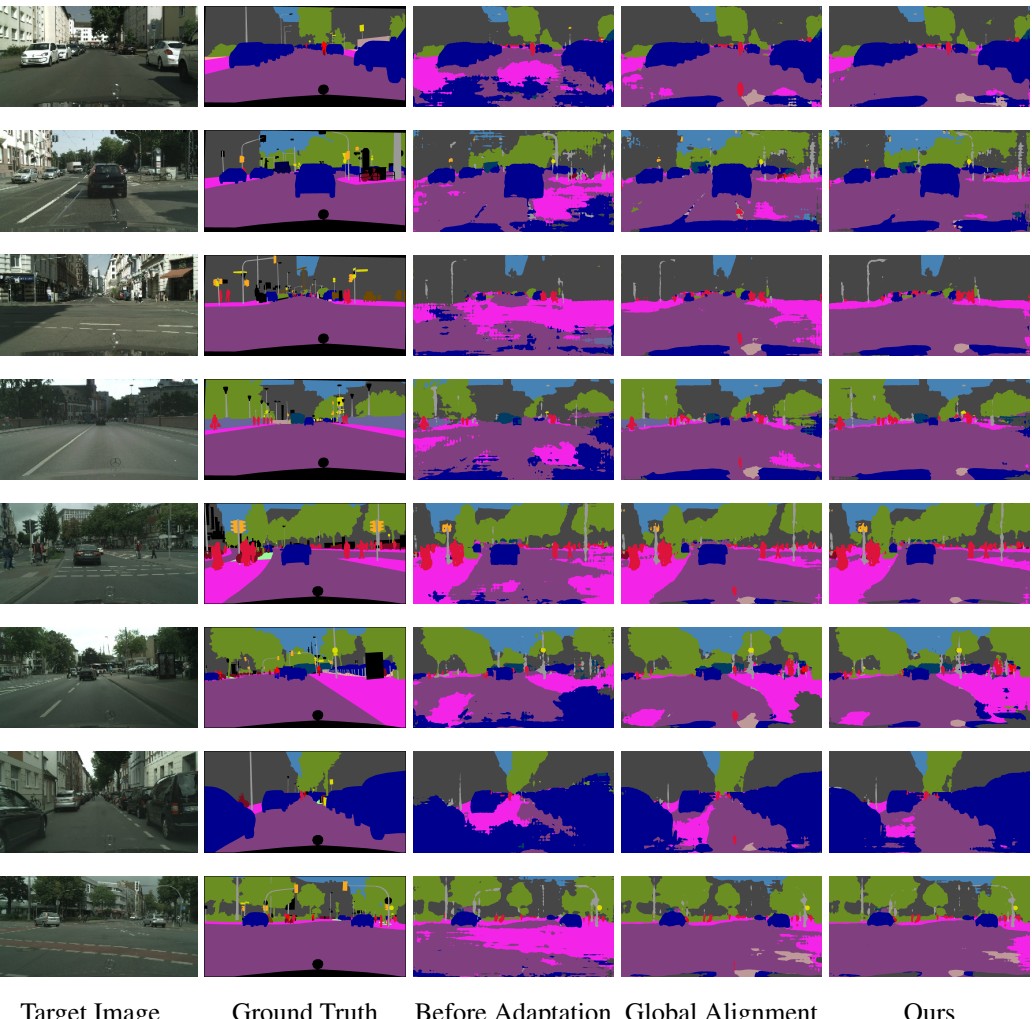

Target Image     Ground Truth     Before Adaptation   Global Alignment      Ours

Figure 11: Example results of adapted segmentation for the SYNTHIA-to-Cityscapes setting. For each target image, we show results before adaptation, output space adaptation Tsai et al. (2018), and the proposed method.

