# OpenReview forum: "Domain Adaptation for Structured Output via Disentangled Patch Representations"
_ICLR.cc/2019/Conference_

### Official Review · AnonReviewer2 · 2018-10-30
**Interesting idea, but mild novelty and missing experiments**

**Rating:** 5
**Confidence:** 4

**Review:**

This paper introduces a domain adaptation approach for structured output data, with a focus here on semantic segmentation. The idea is to model the structure by exploiting image patches, but account for the fact that these patches may be misaligned, and thus not in exact correspondence. This is achieved by defining new patch classes via clustering the source patches according to the semantic information, and making use of an adversarial classifier on the predicted patch-class distributions.

Strengths:
- Modeling the structure via patches is an interesting idea.
- The proposed method achieves good results on standard benchmarks.

Weaknesses:

Method:
- The idea of relying on patches to model the structure is not new. This was achieved by Chen et al., CVPR 2018, "ROAD: Reality Oriented Adaptation...". In this work, however, the patches were assumed to be in correspondence, which leaves some novelty to this submission, although reduced.
- In essence, the patch-based adversarial alignment remains global; this can be thought of as working at a lower resolution and on a different set of classes, defined by the clusters, than the global alignment. The can be observed by comparing Eq. 3 and Eq. 6, which have essentially the same form. This is fine, but was not clear to me until I reached Eq. 6. In fact, what I understood from the beginning of the paper was an alternative formulation, where one would essentially assign each patch to a cluster and aim to align the distributions of the output (original classes) within each cluster. I suggest the authors to clarify this, and possibly discuss the relation with this alternative approach.
- I am not convinced by the claimed relationship to methods that learn disentangled representations. Here, in essence, the authors just perform clustering of the semantic information. This is fine, but I find the connection a bit far-fetched and would suggest dropping it.

Experiments:
- The comparison to the state of the art is fine, but I suggest adding the results of Chen et al, CVPR 2018, which achieves quite close accuracies, but still a bit lower. The work of Saito et al., CVPR 2018, "Maximum Classifier Discrepancy..." also reports results on semantic segmentation and should be mentioned here. I acknowledge however that their results are not as good as the one reported here.
- While I appreciate the ablation study of Section 4.2, it only provides a partial picture. It would be interesting to study the influence of the exact values of the hyper-parameters on the results. These hyper-parameters are not only the weights \lambda_d, \lambda^g_{adv} and \lambda^l_{adv}, but also the number of clusters and the size of the patches used.

Summary:
I would rate this paper as borderline. There is some novelty in the proposed approach, but it is mitigated by the relation to the work of Chen et al., CVPR 2018. The experiments show good results, but a more thorough evaluation of the influence of the hyper-parameters would be useful.

---

> ### Author Response · Authors · 2018-11-11
> **Rebuttal**
>
> Thanks for the valuable comments. For the work in Chen et al., CVPR’18, we acknowledge their idea of using spatial-aware adaptation on spatial regions in the image (will cite it in the revised paper). However, their idea is more similar to the PatchGan discriminator used in Tsai, et al., CVPR’18 (i.e., spatially global alignment), and is different from the proposed patch-level alignment. Our patch alignment focuses on refining small patches (e.g., 32x64) and is location-independent (as described in Figure 1, introduction, and Section 3.3), while the CVPR’18 works assume fixed local grids with larger regions (e.g., 171x342) that account for the context information. In addition, as shown in the ablation study (without reshaped \hat{F} in Table 1), it shows that the proposed location-independent operation helps patch-level alignment.
>
> Although the forms of Eq. 3 and Eq. 6 are similar, they are different in the feature alignment space, where Eq. 6 is guided by the clustering process in the clustered space (for both \hat{F}). As the reviewer mentioned, we do assign a cluster to a source patch for constructing the clustered space, and then we align target patches to this space (also visualized in the appendix C), based on the assumption that source and target patch distributions are shared regardless of where they are in original images. For disentanglement, we will drop this term and instead emphasize the learned discriminative feature representations for patch alignment to reduce the confusion.
>
> For the number of clusters K, we find that the result varies on the GTA5-to-Cityscapes dataset. For example, when K is small (e.g., 20), there would be ambiguities for the patch-level alignment process and the performance drops to 41.6, while it is also more difficult to match patches across domains when K is too large (e.g., 200). In practice, we find that within a reasonable range, e.g., K = [30, 80], the IoU is in a range of [42.6%, 43.2%]. For \lambda_d, the goal is to simply perform classification based on clustering, and we find the results do not differ a lot when choosing \lambda_d from a range of [0.005, 0.02]. For \lambda_adv^g, we directly follow the choice from Tsai, et al., CVPR’18, in which they have provided a study. For \lambda_adv^l in a range of [0.00005, 0.001], the results are in a range of [42.7%, 43.2%]. We will provide a complete analysis in the revised paper. Note that, choosing such hyper-parameters is an open question for domain adaptation tasks. We will put this as a future work and we hope that by providing such analysis, it would help the audience better understand the effect of hyper-parameters.
>
> We will add and compare the mentioned papers in the revised manuscript, including Chen et al., CVPR’18 and Saito et al., CVPR’18.

---

### Official Review · AnonReviewer3 · 2018-11-02
**An interesting idea for disentangled patch representation learning as a drop-in module for UDA. Method effective but relative weak results compared to SOTA**

**Rating:** 5
**Confidence:** 5

**Review:**

This paper proposes a drop-in module of disentangled patch representation learning for adversarial learning-based domain adaptation. The main idea is to encourage the source patch level representation to be disentangled, by creating certain intermediate pseudo-ground truths via clustering the label patch histograms using k-means. This basically creates an alternative, additional view of prediction target of the network outputs. And similar to global network output alignment by Tsai et al., the authors impose an adversarial loss on the additionally introduced view.

Clarity: The paper is well-written with good clarity.

Results: This paper has a good experimental validation of proposed module.

Concerns:
- The idea of using patches in domain adaptation is not completely new. ROAD: Reality Oriented Adaptation for Semantic Segmentation of Urban Scenes, CVPR 2018 also uses the patch level information to help domain adaptation. Although the ideas are not entirely identical, this paper should at least cite and compare this work.

- The disentangled patch feature learning introduces two additional loss, L_d and L_adv^l, which require three extra parameters, including K in K-means, lambda_d and lambda_adv^l. It will be great if a formal sensitivity analysis on the parameters can be conducted. There are some details missing in the paper too. For example, what is the performance of the VGG source model without adaptation? I am also curious about the learning behavior of the proposed method. Could you show the mIoU v.s. epoch curve for GTA2Cityscapes, or any other benchmarks?

- Although consistently improving over Tsai et al., CVPR18, the introduced methods does not show very significant gain in multiple experiments. On SYNTHIA-to-City, only 0.4 mIoU gain is obtained. In addition, while the proposed method is empirically effective, it is largely task-specific and restricted to domain adaptation for scene parsing only. It seems difficult to generalize the same method to other domain adaptation tasks. The limitation on the performance gain and generalizability somehow reduced the contribution from this work to the community.

- A major concern of this work is the lack of citation and direct comparison to multiple previous SOTAs. For example, the paper should compare the end-system performance with several published works such as:
1. Zhang et al., Fully convolutional adaptation networks for semantic segmentation, CVPR2018
2. Zhu et al., Penalizing top performers: conservative loss for semantic segmentation adaptation, ECCV2018
3. Zou et al., Domain adaptation for semantic segmentation via class-balanced self-training, ECCV2018
And according to the results reported by these works, the proposed joint framework in this paper does not seem very competitive in terms of the UDA performance in multiple settings

---

> ### Author Response · Authors · 2018-11-11
> **Rebuttal**
>
> Thanks for the valuable comments. For the CVPR’18 work (ROAD: Reality Oriented Adaptation for Semantic Segmentation of Urban Scenes), we acknowledge their idea of using spatial-aware adaptation on spatial regions in the image (will cite it in the revised paper). However, their idea is more similar to the PatchGan discriminator used in Tsai, et al., CVPR’18 (i.e., spatially global alignment), and is different from the proposed patch-level alignment. Our patch alignment focuses on refining small patches (e.g., 32x64) and is location-independent (as described in Figure 1, introduction, and Section 3.3), while the CVPR’18 works assume fixed local regions with larger patches (e.g., 171x342) that account for the context information. In addition, as shown in the ablation study (without reshaped \hat{F} in Table 1), it shows that the proposed location-independent operation helps patch-level alignment.
>
> For the number of clusters K, we find that the result varies on the GTA5-to-Cityscapes dataset. For example, when K is small (e.g., 20), there would be ambiguities for the patch-level alignment process and the performance drops to 41.6, while it is also more difficult to match patches across domains when K is too large (e.g., 200). In practice, we find that within a reasonable range, e.g., K = [30, 80], the IoU is in a range of [42.6%, 43.2%]. For \lambda_d, the goal is to simply perform classification based on clustering, and we find that the results do not differ a lot when choosing \lambda_d from a range of [0.005, 0.02]. For \lambda_adv^l in a range of [0.00005, 0.001], the results are in a range of [42.7%, 43.2%]. We will provide a complete analysis in the revised paper. Note that, choosing such hyper-parameters is an open question for domain adaptation tasks. We will put this as a future work and we hope that by providing such analysis, it would help the audience better understand the effect of hyper-parameters.
>
> Following Tsai, et al., CVPR’18, the source-only performance using VGG is 26.4% and 30.7% on GTA5-to-Cityscapes and SYNTHIA-to-Cityscapes, respectively. For the mIoU v.s. epoch curve on GTA5-to-Cityscapes, since there is no supervision on the target domain, the performance usually fluctuates as most domain adaptation methods do via adversarial learning. In our experiments, the IoUs are [42.6, 42.0, 42.1, 43.2, 42.1] when training for [50, 55, 60, 65, 70] K iterations using a batch size of 1.
>
> Although the improvement on SYNTHIA-to-Cityscapes is smaller, we find larger gains on certain categories against Tsai, et al., CVPR’18, such as road (3%), sidewalk (2.2%), and sky (1.5%). This is because that the proposed method is designed to overcome domain gaps such as camera pose or field of view via location-independent patch-level alignment. Due to this merit of our approach and the ease of integrating our module into any architectures, we believe that it could be beneficial for other tasks (e.g., depth estimation) that also suffer from the similar issues to semantic segmentation.
>
> We thank for pointing out related works. Different from these methods that mostly focus on the usage of pixel-level domain adaptation (synthesized target images), loss function design, and pseudo label re-training, our work explores a new perspective via patch-level adversarial alignment and the proposed module is general for different architectures or design choices. While the performance is competitive compared to these methods, we believe that our contribution is orthogonal to theirs. We will add and discuss these papers in the revised paper.

---

### Official Review · AnonReviewer1 · 2018-11-02

**Rating:** 7
**Confidence:** 5

**Review:**

The authors tackle the unsupervised domain adaptation problem on tasks with structured output (in this case, semantic segmentation) by performing adversarial alignment at two levels: globally, using the entire image, and locally, using patches of the image. Their global alignment method matches previous adversarial adaptation approaches, so the primary contribution appears to be their patch-level alignment method. They cluster source image patches by histogramming the corresponding label patches, then performing K-means clustering on the histogrammed label features. A new model is trained to reproduce the cluster labels from the source image patches, and this model is adversarially optimized so that target image patches produce a matching feature distribution.

The paper is well-written and concise. It's organized well, and I had very little trouble following the description of their method. The various components of their model are straightforward and well-motivated. They validate their model on multiple synthetic-to-real segmentation tasks, demonstrating strong performance relative to existing baselines, and they also provide a thorough ablation study showing that each of the components of their proposed model is an important part of their final product, which further convinces the reader that the model is sound.

One quibble is that the authors mention disentanglement quite a bit in this paper, including in the title, though it isn't clear to me what is being disentangled. They claim the use of of label information is a disentangling factor, but that seems to be true of domain adaptation approaches in general, which all attempt to disentangle semantic information from domain-specific details in some form or other. Further clarification on precisely what is being disentangled would be helpful.

Another question that lingers is whether or not the additional classification module $H$ and the clustering are truly necessary. A baseline I would like to see would be to remove $H$ entirely and simply train another adversarial discriminator similar to $D_g$ directly on patches of $O$ instead of the full output. This sounds similar to the ablation experiment mentioned in 4.2 where $L_d$ is removed, but my understanding is that ablation experiment still uses an additional featurizer $H$. A more rigorous exploration of the clustering process, such as visualizations of learned clusters and a study of how the number of clusters affects performance would serve to further validate the model.

---

> ### Author Response · Authors · 2018-11-11
> **Rebuttal**
>
> Thanks for the positive and valuable comments. For disentanglement, we agree that most domain adaptation methods utilize sort of a similar idea, while our approach tackles it on the patch-level via a clustering process on patches. To reduce the confusion, we will drop the term of disentanglement and instead emphasize the learned discriminative feature representations for patch alignment.
>
> In the ablation study, the one without $L_d$ is actually the same setting as the reviewer suggested, i.e., removing $H$ entirely and simply training another adversarial discriminator similar to $D_g$ directly on patches. Using an additional featurizer $H$ without $L_d$ would not be valid as there is no supervision from $L_d$. We will clarify this study in the revised paper.
>
> For the number of clusters K, we find that the result varies on the GTA5-to-Cityscapes dataset. For example, when K is small (e.g., 20), there would be ambiguities for the patch-level alignment process and the performance drops to 41.6, while it is also more difficult to match patches across domains when K is too large (e.g., 200). In practice, we find that within a reasonable range, e.g., K = [30, 80], the IoU is in a range of [42.6, 43.2]. Note that, we do not really tune this K for different datasets but use the same K=50 for all the scenarios. In the appendix C, we show some examples of clusters and visualizations for patch alignment.

---

### Public Comment · (anonymous) · 2018-10-27
**Ablation study for K-means**

Hi, authors
I have a question about the K-means. You use K=50 in all your experiments, and have you try other value of K?
I think that the K may vary with different datasets due to the different appearance distributions. An ablation study will make it clear.

And there are some segmentation adaptation works missing in the related  work section.
Conditional Generative Adversarial Network for Structured Domain Adaptation, CVPR2018
Fully Convolutional Adaptation Networks for Semantic Segmentation, CVPR2018
Penalizing Top Performers: Conservative Loss for Semantic Segmentation Adaptation, ECCV2018
DCAN: Dual Channel-wise Alignment Networks for Unsupervised Scene Adaptation, ECCV2018
Unsupervised Domain Adaptation for Semantic Segmentation via Class-Balanced Self-Training, ECCV2018

---

> ### Author Response · Authors · 2018-10-29
> **Ablation study for the value K and related work**
>
> Thanks for the comments. For the value of K, we find that the result varies on the GTA5-to-Cityscapes dataset. For example, when K is small (e.g., 20), there would be ambiguities for the patch-level alignment process and the performance drops to 41.6, while it is also more difficult to match patches across domains when K is too large (e.g., 200). In practice, we find that within a reasonable range, e.g., K = [30, 80], the IoU is in a range of [42.6, 43.2]. Note that, we do not really tune this K for different datasets but use the same K=50 for all the scenarios.
>
> We appreciate the above-mentioned works for semantic segmentation adaptation. Different from these works that mostly focus on the usage of pixel-level domain adaptation (synthesized target images), loss function design, and pseudo label re-training, our work explores a new perspective via patch-level adversarial alignment for structured output, and the proposed module is general for different architectures or design choices. We will add and discuss these papers in our manuscript.

---

### Public Comment · (anonymous) · 2018-11-11
**The ResNet-101 network may not be pre-trained on Image-Net**

In section 3.5, you mention that you follow the framework used in (Tsai et al., 2018) and the ResNet-101 is pre-trained on ImageNet. However, it seems that the network used in (Tsai et al., 2018) is pre-trained on COCO dataset. The author mentioned it in here. (https://github.com/wasidennis/AdaptSegNet/issues/5)

Learning to Adapt Structured Output Space for Semantic Segmentation Y.-H. Tsai and W.-C. Hung and S. Schulter and K. Sohn and M.-H. Yang and M. Chandraker CVPR 2018

---

> ### Author Response · Authors · 2018-11-12
> **Pre-trained model**
>
> Thanks for pointing this out. Yes, we have found this issue and used the pre-trained model on ImageNet only for ResNet-101.

---

> > ### Public Comment · (anonymous) · 2018-11-13
> > **Pre-trained model**
> >
> > As you reported on the Table2, you can get 36.6%mIoU in source only setting, which is the same result reported on Tsai's paper. However, I have attempted to replace the COCO pre-trained model to Image-Net and use Tsai 's code to train the adaptation task, I only can get 21.1% mIoU and 29.9% mIoU for their source only and output Space approach.
> >
> > Moreover, I have not found any paper reported that they can get 35%+ source only mIoU result trained on ResNet-101 Image-Net pre-trained model. I am wondering that what things you have modified on Tsai's code. And if it is possible, do you mind that open source the code for study after review?
> >
> > Fully Convolutional Adaptation Networks for Semantic Segmentation Yiheng Zhang, Zhaofan Qiu, Ting Yao, Dong Liu, Tao Mei, CVPR 2018
> > ROAD: Reality Oriented Adaptation for Semantic Segmentation of Urban Scenes Yuhua Chen, Wen Li, Luc Van Gool CVPR 2018

---

> > > ### Author Response · Authors · 2018-11-13
> > > **Pre-trained model**
> > >
> > > For the base model we used in our paper, we have not revised the released code from Tsai, et al., CVPR'18. But, while loading the pre-trained weights from ImageNet, it needs to be careful as those weights are originally imported from the Caffe framework.
> > >
> > > Yes, we directly took the 36.6% mIoU from Tsai, et al., CVPR'18, as the same base model is used in our paper. We also used a VGG-16 version with ImageNet pre-training, and can reach 26.4% mIoU on GTA5-to-Cityscapes for the source-only model.
> > >
> > > We would be happy to release our code/model after the review process. At this point, we suggest you to directly report your case on their Github page for further assistant.

---

### Public Comment · (anonymous) · 2019-01-26
**Related work on local domain adaptation approaches**

There are some local domain adaptation based approaches that can be cited -

[1] Courty, Nicolas, et al. "Optimal transport for domain adaptation." IEEE transactions on pattern analysis and machine intelligence 39.9 (2017): 1853-1865.

[2] Das, Debasmit, and CS George Lee. "Sample-to-sample correspondence for unsupervised domain adaptation." Engineering Applications of Artificial Intelligence 73 (2018): 80-91.

[3] Debasmit Das and C.S. George Lee, “Unsupervised Domain Adaptation Using Regularized Hyper-Graph Matching,” Proceedings of 2018 IEEE International Conference on Image Processing (ICIP), Athens, Greece, pp. 3758-3762, October 7-10, 2018.

[4] Debasmit Das and CS George Lee. “Graph Matching and Pseudo-Label Guided Deep Unsupervised Domain Adaptation,” Proceedings of 2018 International Conference on Artificial Neural Networks (ICANN), Rhodes, Greece, pp. 342-352, October 4-7, 2018.

---

### Meta-Review · Area_Chair1 · 2018-12-12
**An attempt at incorporating structure from output into an adaptation pipeline**

**Confidence:** 4
**Recommendation:** Reject

**Metareview:**

The paper explores unsupervised domain adaptation when the output is structured. Here they focus experimentally on semantic segmentation in driving scenes and use the spatial structure of the scene to produce two losses for adaptation: one global and one patch based. The method tackles an important problem and proposes a first attempt at a new solution. While the the experiments are missing ablations and some comparisons to prior work as noted by the reviewers, the authors have provided comments in their rebuttal explaining the relation to the prior work and promising to include more in the revised manuscript.

The paper is borderline, but falls short on the necessary updates requested by reviewers.  The use of the structured output which is available in semantic segmentation of driving scenes is a useful direction. The paper is missing enough key results and analysis in it's current form to be accepted.